# Bioactivity-Guided Fractionation and Identification of Antidiabetic Compound of *Syzygium polyanthum* (Wight.)’s Leaf Extract in Streptozotocin-Induced Diabetic Rat Model

**DOI:** 10.3390/molecules27206814

**Published:** 2022-10-12

**Authors:** Tri Widyawati, Nor Adlin Yusoff, Idris Bello, Mohd Zaini Asmawi, Mariam Ahmad

**Affiliations:** 1Pharmacology and Therapeutics Department, Medical Faculty, Universitas Sumatera Utara, Medan 20155, Indonesia; 2Department of Pharmacology, School of Pharmaceutical Sciences, Universiti Sains Malaysia, Penang 11800, Malaysia; 3Department of Toxicology, Advanced Medical and Dental Institute, Sains@Bertam, Universiti Sains Malaysia, Penang 13200, Malaysia

**Keywords:** antihyperglycemic, Indonesian bay leaves, *Syzygium polyanthum* (Wight.), diabetes, squalene

## Abstract

(1) Background: An earlier study on the hypoglycemic activity of *S. polyanthum* (Wight.) leaf methanol extract identified squalene as the major chemical compound. The present study was conducted to assess the hypoglycemic effect of fractions and subfractions of the methanol extract of *S. polyanthum* compared to the squalene using a bioassay-guided in vivo study. (2) Methods: The methanol extract was fractionated using the liquid–liquid fractionation method. Streptozotocin-induced type 1 diabetic rat was used to study the hypoglycemic effect. (3) Results: The findings showed that chloroform fraction significantly (*p* < 0.05) lowered blood glucose levels of diabetic rats as compared to the control. Further fractionation of chloroform fraction yielded subfraction-1 and -2, whereby subfraction-1 exhibited a higher blood-glucose-lowering effect. The lipid profile test showed that the total cholesterol level of subfraction-1 and squalene-treated groups decreased significantly (*p* < 0.05). An immunohistochemistry study revealed that none of the treatments regenerated pancreatic β-cells. Gas chromatography–mass spectrophotometer analysis identified the presence of squalene in the active methanol extract, chloroform fraction, and subfraction-1. In silico analysis revealed a higher affinity of squalene against protein receptors that control lipid metabolism than metformin. (4) Conclusions: Data obtained from the present work suggested the crude methanol extract exerted the highest hypoglycemic effect compared to fraction, subfraction, and squalene, confirming synergistic effect may be responsible for the hypoglycemic activity of *S. polyanthum*.

## 1. Introduction

Diabetes mellitus is a chronic metabolic disorder characterized by persistent hyperglycemia resulting from insufficiency in insulin action or insulin secretion, or both [1]. Persistent hyperglycemia can lead to macrovascular and microvascular complications that affect several vital organs, including the kidneys, eyes, and heart [2]. The majority of diabetes cases are divided into type 1 diabetes mellitus or insulin-dependent and type 2 diabetes mellitus or noninsulin-dependent [3]. The prevalence of diabetes has been steadily increasing over the decades. In 2021, approximately 537 million diabetes cases were reported, and the disorder was the ninth leading cause of death worldwide [4]. Currently, oral hypoglycemic agents and insulin have been used to control the disease and its complications. Several associated adverse effects, however, have been reported, reflecting the need for continuous research to develop new effective therapeutic agents for diabetes mellitus.

As plants are known to contain a wide range of active compounds, many studies investigate plant sources for potential antidiabetic agents. Plant-derived active compounds serve as pure drugs, as well as lead compounds, for the development of synthetic drugs directly or indirectly [5]. Bioassay-guided fractionation is one of the widely used techniques in plant drug discovery. In this technique, the extraction and biological screening of the extract take place simultaneously in order to identify the active compound [6]. By applying this technique, several studies have successfully isolated compounds with antihyperglycemic effects, such as xyloccensin-I from *Xylocarpus granatum* [7], nicotiflorin and tulipanin from *Punica granatum* var. [8], and a novel decahydro-1H-xanthene from *Garcinia cowa* [9].

*Syzygium polyanthum* (Wight.) Walp. (Myrtaceae) is an herb widely used in Indonesian and Malaysian cuisines [10]. It is also commonly used as a traditional medicine to treat diabetic patients in Indonesia [11]. Our previous study demonstrated the antidiabetic activity of an *S. polyanthum* methanolic extract in a streptozotocin-induced diabetic rat model, in which squalene was identified as one of the active constituents [12]. Squalene has been reported to improve the lipid profile associated with the progression of diabetes. Gabas-Rivera et al. [12] reported that dietary squalene increased the levels of HDL cholesterol and paraoxonase-1 and decreased total cholesterol levels at a 1 g/kg dose in Apoe-deficient mice. Similarly, Liu et al. [13] showed that the intake of squalene dietary supplements effectively increased HDL levels in obese/diabetic KK-A^y^ mice. These findings were further confirmed by Mirmiranpour et al. [14], who studied the effect of squalene on type 2 diabetic patients, and found that consuming squalene for 84 days caused HDL levels to rise, subsequently lowering total cholesterol, LDL, and VLDL levels in diabetic patients.

The present study utilized a bioactivity-guided approach to further investigate the hypoglycemic activity of *S. polyanthum* leaves and the squalene isolated from the plant. A streptozotocin-induced diabetic rat model that mimicked type 1 diabetes mellitus in humans was used to screen for possible hypoglycemic effects.

## 2. Results and Discussion

### 2.1. Gas Chromatography–Mass Spectrometry Analysis

Gas chromatography–mass spectrometry analysis was conducted to identify the presence of bioactive compound/s in the active extract (methanol extract), the fraction (chloroform fraction), and the subfraction (subfraction-1). Table 1 lists the detected compounds of high library matching quality. The analysis revealed the presence of several bioactive compounds that may contribute to the observed glucose-lowering effect; one compound was found in methanol extract, five in chloroform fraction, and three in subfraction-1. Interestingly, only squalene was detected in all samples. Squalene, a triterpene, is an isoprenoid compound that belongs to the terpenoid family [15]. It has been implicated in several studies as a compound that contributes to the hypoglycemic and anti-obesity activities of plants. Wang et al. [16] have demonstrated the excellent antidiabetic of *Sanbai* melon seed oil with squalene as one of the major bioactive compounds. Ravi Kumar et al. [17] reported that squalene modulated the metabolism of fatty acids in obese diabetic mice models, thus contributing to the glucose-lowering effect. 

### 2.2. Effect on Blood Glucose Level 

Active methanol extract of *S. polyanthum* was fractionated into chloroform, ethyl acetate, *n*-butanol, and water fractions. The result showed all the fractions reduced blood glucose levels after repeated oral administration with a significant lowering effect only seen in the chloroform (*p* < 0.05) fraction-treated group, as compared to the diabetic control group (Figure 1). Hence, the chloroform fraction was further fractionated. Subfractionation of 10 g of chloroform fraction produced 63% of subfraction-1 and 14% of subfraction-2. Only subfraction-1 significantly reduced blood glucose level (*p* < 0.05) when compared with the diabetic control (Figure 2). Overall, the results of this study pertaining to the hypoglycemic activity of *S. polyanthum* extract were consistent with the previous studies, which demonstrated the ability of ethanolic extract of the leaves to substantially reduce the blood glucose levels of alloxan-induced diabetic rats [18,19].

Figure 3 presents the blood glucose level of active extract, fraction, subfraction, squalene, as well as controls in streptozotocin-induced diabetic rats. Compared to the normal rats, streptozotocin induction caused a 2–3-fold increase in blood glucose levels. Blood glucose levels of methanol extract, chloroform fraction, subfraction-1, squalene, and metformin-treated groups were significantly decreased on day 12 as compared to day 0. The descending order of the hypoglycemic activity with reduction percentage was as follows: metformin = methanol extract (56%) > chloroform fraction = subfraction-1 = squalene (43%). Interestingly, the blood-glucose-lowering effect of methanol extract was comparable to that observed in the metformin-treated groups (*p* < 0.001). Overall, crude extract exerted a better hypoglycemic effect than fraction and subfraction. As can be seen in Figure 3, the magnitude of the hypoglycemic effect decreased when the extract was further fractionated. As compared with the NC, only the ME group showed an insignificant difference, which further indicated treatment with ME successfully eliminated hyperglycemia in the diabetic rats. In addition to that, the analysis also suggested the possible synergistic effect between squalene and other bioactive compounds of the methanol extract, chloroform fraction, and subfraction-1. Compared to the pharmacological effects observed with individual isolated compounds, better results are obtained with whole plant extracts due to the presence of various bioactive compounds that work in synergy by targeting either the same or different pathways [20].

In reference to the gas chromatography–mass spectrometry analysis (Table 1), five compounds were detected in chloroform fraction, three in subfraction-1, and one compound from methanol extract. Additionally, the chloroform fraction has a higher percentage of squalene (8.92%) than methanol extract (7.60%) and subfraction-1 (4.54%). The number of detected compounds and percentage of squalene, however, do not reflect the magnitude of the observed hypoglycemic effect of each sample. This further implies the presence of other potentially bioactive compounds in the methanol extract that are not detected through gas chromatography–mass spectrophotometry analysis. Even though gas chromatography–mass spectrophotometry is considered the gold standard for a broad spectrum of compound screening, it is unable to directly analyze compounds that are polar and nonvolatile [21]. Chemical derivatization to create volatile forms of the compounds is needed to make them amenable for this chromatographic analysis. 

### 2.3. Effect on Body Weight

Figure 4 shows the effect of *S. polyanthum* extract, fraction, and subfraction on body weight. Normal rats showed a significant increase in body weight after the 12-day experimental period (*p <* 0.01). As expected, diabetic control rats exerted a significant loss in body weight (*p <* 0.05) on day 12. The significant loss of body weight in untreated diabetic rats may be due to the increased muscle wasting and loss of muscle mass and adipose tissue caused by the excessive breakdown of tissue proteins and fatty acids [22,23]. Several studies have reported similar significant weight reductions in untreated diabetic rats [24,25,26]. Diabetic rats treated with methanol extract, chloroform fraction, subfraction-1, and squalene, as well as metformin, also showed a reduction in body weight after 12 days of the experiment. The reduction, however, was not significant as compared to pre-treatment levels on day 0. The results suggested that the given treatments managed to prevent further significant weight loss in diabetic rats. The minimal weight loss of treated groups was due to the gradual decrease in blood glucose levels. Additionally, the effect of the treatments on body weight might be more prominent if the diabetic rats were treated for a longer duration. Taher et al. [27] and Azad and Sulaiman [28] reported a significant increase in body weight of treated diabetic rats only after 14 days of treatments.

### 2.4. Effect on Lipid Profile

Figure 5 presents the effects of *S. polyanthum* on the lipid profiles of streptozotocin-induced diabetic rats. As compared with the diabetic control, only subfraction-1 and squalene showed the ability to decrease total cholesterol levels significantly (*p <* 0.05), an effect which was also observed in the metformin-treated group (*p <* 0.05). Metformin and squalene-treated groups also significantly reduced low-density lipoprotein levels (*p* < 0.001 and *p* < 0.05 respectively). Those findings confirmed the promising hyperlipidemic effect of squalene, as reported by previous studies. Liu et al. [29] showed that a high squalene diet significantly reduced triglycerides and cholesterol levels in rats. Several years later, Gabas-Rivera et al. [30] further suggested that squalene managed to reduce cholesterol levels in a dose-dependent manner. 

Non-significant changes in lipid parameters were seen in diabetic rats treated with methanol extract and chloroform fraction. Diabetes was shown to be associated with hyperlipidemia [31], and optimal glycemic control ameliorates lipid profile abnormalities [32]. However, in the author’s study, normalization of plasma glucose levels in methanol extract and chloroform fraction treated groups did not restore the lipid parameters, implying that factors, in addition to glycemic control, are also involved. An example of this factor is insulin. Insulin has been reported to prevent hypercholesterolemia in type 1 diabetes. Insulin, by suppressing the hepatic FoxO1 gene, lowered 12α-hydroxylated bile acids, plasma cholesterol, and cholesterol absorption [33]. This means the cholesterol levels remain high in the deficiency of insulin.

### 2.5. Insulin Level and Immunohistochemistry Assessment

Figure 6 shows the immunohistochemically stained pancreatic tissues under 40 × 10 magnification power. The brown-colored area represents insulin in the viable β-cells. In the Islets of Langerhans of the normal rat, clear and large brownish spots were seen. The β-cells were found to make up 87.92 ± 1.9% of Islets of Langerhans in the normal rats. In the streptozotocin-induced diabetic rats, the size of the brown-colored areas was significantly (*p* < 0.001) reduced and comprised 21.36 ± 3.7% of the total islets of the Langerhans area (Figure 7A). With the breaking of the DNA strand, interruption of the glucose transport, and interference with the function of glucokinase, the streptozotocin damaged the β-cells [34]. All the treated groups showed a tendency to have larger immunostained areas. The effect, however, was insignificant when compared with the diabetic control group. This indicates that the treatments were ineffective in terms of regenerating and protecting viable pancreatic β-cells. Figure 7B depicts the concentration of serum insulin in the treated groups. As expected, the serum insulin levels in all diabetic rats were three-fold less than those of the normal control (*p* < 0.001). Only subfraction-1 significantly increased the level of serum insulin as compared to the diabetic controls (*p <* 0.05). This finding explained the positive effect of subfraction-1 in lowering cholesterol levels and confirmed the role of insulin in normalizing cholesterol levels. Semova et al. [33], who studied the role of insulin regulation on the cholesterol mechanism in mouse model type 1 and humans with type 1 diabetes, have suggested that insulin modulated the level of plasma cholesterol via inhibition of hepatic FoXO1, which led to the reduction of 12α-hydroxylated bile acids, cholesterol absorption, and plasma cholesterol levels.

Considering diabetes mellitus is a multifactorial chronic disease, further research is needed to better understand the possible mechanism of the hypoglycemic action of this plant by exploring the effect of *S. polyanthum* on the hepatic glucose production, intestinal glucose absorption, and peripheral glucose uptake in the muscle and adipose tissues. Additionally, the current study has limitations in terms of the duration of the treatment. Effects of the treatment on the parameters such as body weight and lipid profile can be more profound in a longer study period.

### 2.6. Molecular Docking Analysis 

Using iGEMDOCK, a molecular docking simulation was carried out to determine the effectiveness of squalene’s binding to the chosen diabetes proteins. Metformin was applied for comparison purposes. Figure 8 shows the binding energy of all two ligands on the 13 protein targets. Based on the analysis of binding energy, the lowest negative scores of total energy indicate a higher affinity between the ligand and protein’s receptors, which further signifies that the ligands could modulate or inhibit the respective protein target. The binding energy of squalene to all the target proteins was found to be lower than that of the standard drugs, metformin, except for 4Y14, 1XU7, and 1IR3. Squalene showed the highest affinity against 2Q5S (−75.9 kcal/mol), followed by 2HWQ (−71 kcal/mol), 1ZON (−65.56 kcal/mol), and 1FM9 (−64.28 kcal/mol). Receptors 2Q5S, 2HWQ and 1FM9 are the proteins that regulate and control adipogenesis, energy balance, and biosynthesis of lipids [35,36,37]. This in silico analysis is in agreement with the findings of Mirmiranpour et al. [14] on the effect of squalene on lipid profiles in type 2 diabetic patients. Further study is needed to predict the active site and associated amino acids of target diabetic proteins.

## 3. Materials and Methods

### 3.1. Chemicals

Metformin 500 mg, a standard oral antidiabetic drug, was used as the positive control. Streptozotocin and squalene (CID 638072) were purchased from Sigma-Aldrich Chemical Company (St. Louis, MO, USA). All reagents and chemicals used are analytical grades.

### 3.2. Plant Collection and Identification

*Syzygium polyanthum* leaves were collected from Titi Kuning, Medan, Indonesia (Geographical coordinates: 3.522988, 98.682834) from July to October 2011. The plant was identified by Dr. Nursahara Pasarbibu from the School of Biological Sciences, Bioteknologi no.1 Kampus USU, University of Sumatera Utara, Medan, Indonesia (voucher specimen number: no.13/MEDA/2012). 

### 3.3. Preparation of Samples

The dried leaves were powdered using a milling machine. To obtain the extracts of *S. polyanthum*’s, about 1.5 kg of the powder was sequentially macerated (40 °C–60 °C) with 4.9 L each of the respective solvents: petroleum ether, chloroform, and methanol. The extracts were filtered using Whatman No.1 filter paper and concentrated *in vacuo* by using a rotary evaporator (Labortechnik, AG CH-9230 Flawil, Switzerland) at reduced pressure. The concentrated extracts were dried in an oven (40 °C) until traces of the organic solvent completely evaporated. An earlier study has shown that methanol extract exerted the most potent hypoglycemic effect [12]. Hence, in this study, the active methanol extract was selected for further bioassay-guided fractionation.

The active methanol extract (25 g) was sequentially fractionated by a liquid–liquid partition with four solvents, chloroform (250 mL), ethyl acetate (250 mL), *n*-butanol (250 mL), and water. This resulted in four fractions of chloroform, ethyl acetate, *n*-butanol, and water. The water fraction underwent lyophilization using a freeze dryer (Labonco Corporation, Kansas, MO, USA). The preliminary result showed that chloroform fraction significantly lowered the blood glucose level of streptozotocin-induced diabetic rats as compared to the control. Thus, the most active fraction, the chloroform fraction, was further fractionated by dissolving 0.5 g in *n*-hexane (100 mL). Meanwhile, the *n*-hexane was drained and replenished until no color had formed. The *n*-hexane fraction was filtered, concentrated by a rotary evaporator, and freeze-dried to obtain subfraction 1. The remaining residue, which was dissolved in chloroform, was also filtered. The chloroform portion was concentrated in vacuo to yield subfraction 2. Figure 9 depicts the schematic extraction and fractionation of *S. polyanthum* leaves. All extracts, fractions, and subfractions were kept in a freezer (−20 °C) until further use. Doses were freshly prepared using 5% Tween 80 in 0.9% normal saline prior to oral administration.

### 3.4. Gas Chromatography–Mass Spectrometry (GC–MS) Analysis

For GC–MS analysis, 4 mg of each active methanol extract, chloroform fraction, and subfraction-1 were respectively dissolved in 1 mL of chloroform. The reference compound, squalene (410.72 g/mol, >98% liquid), was diluted in methanol at the concentration of 1 mg/mL. The presence of squalene in methanol extract, chloroform fraction, and subfraction-1 was confirmed by using ion fractionation. The percentage of squalene was calculated based on the peak heights in methanol extract, chloroform fraction, and subfraction-1 respective spectra.

### 3.5. In Vivo Antihyperglycemic Studies

#### 3.5.1. Induction of Type 1 Diabetes Mellitus in Rats

Healthy male *Sprague Dawley* rats weighing between 200 and 250 g were obtained from the Animal Research and Service Centre, Universiti Sains Malaysia, Penang, Malaysia. Before commencement of the study, animals were acclimatized in the Animal Transit Room, School of Pharmaceutical Sciences, Universiti Sains Malaysia, for one week at 20–22 °C with a 12 h light/dark cycle.

Freshly prepared streptozotocin in 0.9% sodium chloride was injected intraperitoneally at a dose of 55 mg/kg to 16-h fasted rats. The rats were provided with 10% dextrose drinking water for the first 24 h to avoid the incidence of fatal hypoglycemia. After 72 h, the blood glucose levels of the fasting animals were measured, and animals with blood glucose levels above 200 mg/dL (11 mmol/L) were selected for the study.

#### 3.5.2. Bioassay-Guided Antihyperglycemic Activity of *S. polyanthum*

Hypoglycemic activity of *S. polyanthum*’s fractions and subfractions were assessed using streptozotocin-induced diabetic rats. Diabetic rats were randomly divided into groups of six (*n* = 6). Normal saline (10 mL/kg BW) was given and acted as a negative control group, and metformin at the dose of 500 mg/kg BW was used as the positive control. To screen the hypoglycemic activity of fractions, diabetic rats were treated with 500 mg/kg of chloroform, ethyl acetate, *n*-butanol, and water fractions, respectively. All treatments were administered orally, twice daily, for 6 days. Blood glucose levels were measured before and after the treatment. The same step was repeated to evaluate the hypoglycemic activity of subfractions, subfraction-1, and subfraction-2. These subfractions, at the dose of 250 mg/kg BW, were given orally, and blood glucose levels before and after treatment were measured. 

Finally, once the active fraction and subfraction have been determined, the hypoglycemic activity of the active extract, fraction, and subfraction, as well as squalene, was conducted again at the same time to avoid time period bias. The procedure is summarized in Figure 10. The treated groups were as follows: Four groups of diabetic rats (*n* = 6) were respectively treated with methanol extract (1 g/kg B.W.), chloroform fraction (500 mg/kg B.W.), subfraction-1 (250 mg/kg B.W.), and squalene (160 mg/kg B.W.). The fifth and sixth groups received metformin (500 mg/kg B.W.) and normal saline (10 mL/kg B.W.) and served as respective positive control and diabetic control. Six normoglycemic rats were treated with saline (10 mL/kg) and served as the normal control. The minimum effective dose of squalene was determined previously, as shown in Appendix A. All treatments were administered orally twice daily for 12 days. After a 12-day treatment, the rats were euthanized using carbogen (95% CO_2_ and 5% O_2_), and a blood sample (3 mL) was obtained via cardiac puncture and centrifuged at 3000 rpm for 10 min to collect the serum. The serum was stored at −20 °C until biochemical parameters analysis. The pancreas was harvested for immunohistochemistry analysis.

#### 3.5.3. Measurement of Fasting Blood Glucose

Blood glucose level was measured at two interval points, day 0 (before treatment) and day 12 (after repeated treatment). The rats were fasted overnight before blood glucose measurement. Approximately fifty microliters of blood was obtained from the tail vein of each rat, and blood glucose level was determined using Accu-Check Advantage Clinical Glucose Meter (Roche Diagnostics Co., Indianapolis, IN, USA).

#### 3.5.4. Measurement of Serum Insulin Level, Lipid Profile, and Body Weight

The concentration of insulin in the serum was determined in triplicates using an ultra-sensitive rat insulin ELISA kit (Crystal Chem Inc, Elk Grove Village, IL, USA). Total cholesterol, triglycerides, and HDL cholesterol were determined using an automated Siemens ADVIA 2400 Chemistry Analyzer (Erlangen, Germany). Total cholesterol and triglycerides were measured based on the enzyme-coupled reaction, whereas HDL cholesterol was determined using a direct HDL-cholesterol assay. LDL cholesterol was calculated using the Friedewald equation. The body weights of the rats were measured on day 0 and day 12 by using an electronic balance (Navigator^TM^, Ohaus Corporation, Nanikon, Switzerland).

#### 3.5.5. Immunohistochemistry Study of Pancreas

A histological assessment of the pancreas was conducted according to the immunohistochemical method described by Yusoff et al. [38]. The pancreatic tissues were fixed in 10% buffered formalin, processed using a tissue processor, and embedded in paraffin. The paraffin-embedded tissues were sectioned into 5 μm slices and mounted on a poly-L-lysine-coated microscope slide. Immunohistochemically staining was performed using a guinea-pig polyclonal antibody of rat insulin. The sections were treated with 3% hydrogen peroxide in methanol to quench endogenous peroxidases, followed by washing in buffer. Then, the sections were incubated in a diluted normal serum for 20 min. The primary antibody, guinea-pig polyclonal insulin antibody, was applied for 30 min, followed by buffer washing for 5 min. The sections were incubated for 30 min with the biotinylated secondary antibody, followed by washing. Following a further 30-min incubation period in a Vectastain ABC kit, washing was performed for 5 min before adding diaminobenzidine for 3–5 min. The sections were slightly counterstained with Harris Hematoxylin, dehydrated, cleared, and mounted. The pancreatic islets were examined under a light microscope (Leica^®^ DMi1, Leica Microsystems, Wetzlar, Hesse, Germany). An image analyzer (Leica^®^ microsystem Qwin plus) was used to analyze the digital image and calculate the percentage (%) of the insulin-containing area of β cells.

### 3.6. Molecular Docking Analysis

Selection of protein targets was conducted by referring to Rao et al. [39]; 3D structures of 13 proteins that are vitally important in diabetes mellitus were retrieved from the Research Collaboratory for Structural Bioinformatics (RCSB) Protein Data Bank (https://www.rcsb.org/structure/2Q5S), accessed on 20 September 2022 and saved in PDB format. Proteins with PDB ID: 1FM9 (Heterodimer of the human retinoic acid receptor α and peroxisome proliferator-activated receptor γ ligand binding domains bound with 9-cis retinoic acid and GI262570 and co-activator peptides), 1IR3 (Phosphorylated insulin receptor tyrosine kinase in complex with peptide substrate and ATP analog), 1XU7 (Tetrameric 11B-HSD1), 1ZON (CD11A I-domain without bound cation), 2HWQ (Peroxisome proliferator-activated receptor agonists), 2Q5S (Peroxisome proliferator-activated receptor γ bound to partial agonist NTZDPA), 2QMJ (N-terminal subunit of human maltase-glucoamylase in complex with acarbose), 2ZJ3 (Isomerase domain of human glucose:fructose-6-phosphate amidotransferase), 3C45 (Human dipeptidyl peptidase IV/CD26 in complex with a fluoroolefin inhibitor), 3CTT (N-terminal human maltase-glucoamylase with casuarine), 3L2M (Pig pancreatic alpha-amylase with alpha-cyclodextrin), 4A5S (Human DPP4 in complex with a novel heterocyclic DPP4 inhibitor), and 4Y14 (Tyrosine phosphatase 1B complexed with inhibitor) were employed in focused molecular docking investigations on diabetes receptor proteins. Preparation of ligand was conducted applying canonical SMILES format of the compounds. Canonical SMILES of ligand squalene was retrieved from the PubChem compound database (https://pubchem.ncbi.nlm.nih.gov/, accessed on 20 September 2022) with PubChem CID: 638072, and the three-dimensional structure of the molecule was simulated using the online server NovoPro Bioscience Inc., Shanghai, China (https://www.novoprolabs.com/tools/smiles2pdb, accessed on 20 September 2022) and was saved in PDB format. Comparative analysis of the ligand was conducted against metformin drug obtained from PubChem database metformin (PubChem CID:4091). The 2D structures of squalene and metformin were generated using Novoprolabs (Figure 8). Molecular docking between the ligand and the receptors was carried out using the Generic Evolutionary Method for molecular DOCKing (iGEMDOCKv4.2). iGEMDOCK employs generic evolutionary algorithms (flexible docking method). Docking each ligand to the 13 target proteins was carried out at the standard docking option of 70 generations, 200 population size, and 2 solutions. Bond energies of squalene were compared to those of the standard drug metformin.

### 3.7. Statistical Analysis

The data were expressed as mean ± standard error of the mean (SEM). Statistical significance was determined by Graphpad prism version 7. One-way ANOVA was used, followed by Dunnet’s post hoc test. Pre-treatment and post-treatment comparisons were performed using the paired *t*-test. Differences were considered significant, with the *p*-value being less than 0.05.

## 4. Conclusions

In conclusion, the present work successfully establishes the hypoglycemic activity of *S. polyanthum* leaf extract, demonstrating evidence that *S. polyanthum* is the most effective antidiabetic plant when administered as a whole extract rather than fractions. This suggests that the hypoglycemic effect is a product of the synergy of several compounds in the plant rather than a few or a single compound therein. In addition to that, the observed hypoglycemic effect has no association with enhanced insulin secretion by β-cells. These warrant future studies to explore the possible mechanism of the hypoglycemic effect of *S. polyanthum* and squalene at the extrapancreatic level. 

## Figures and Tables

**Figure 1 molecules-27-06814-f001:**
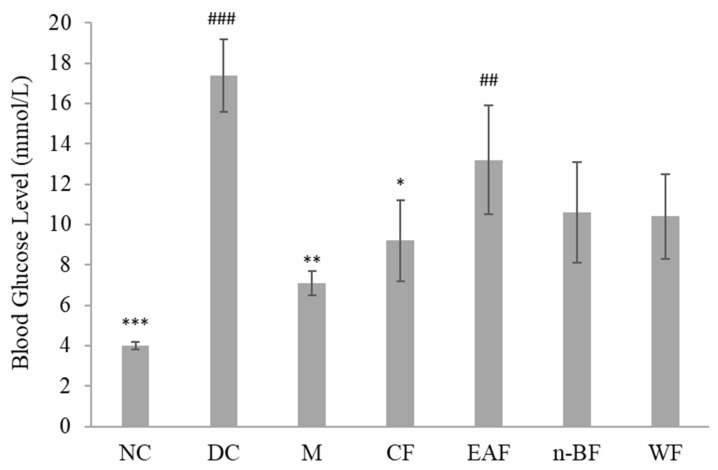
Effects of the fractions of *S. polyanthum*’s methanol extract on the blood glucose levels of streptozotocin-induced diabetic rats after being treated orally twice daily for six days. NC, normal control (normal saline, 10 mL/kg); DC, diabetic control (normal saline, 10 mL/kg); M, metformin (500 mg/kg); CF, chloroform fraction (500 mg/kg); EAF, ethyl acetate fraction; *n*-BF, *n*-butanol fraction (500 mg/kg); WF, water fraction (500 mg/kg). The values are expressed as mean ± SEM (*n* = 6). # indicates significant differences as compared to the NC (^###^
*p* < 0.001 and ^##^
*p* < 0.01). * indicates significant differences compared to the DC (*** *p <* 0.001, ** *p <* 0.01, and * *p* < 0.05) as analyzed using Dunnett’s as post hoc test.

**Figure 2 molecules-27-06814-f002:**
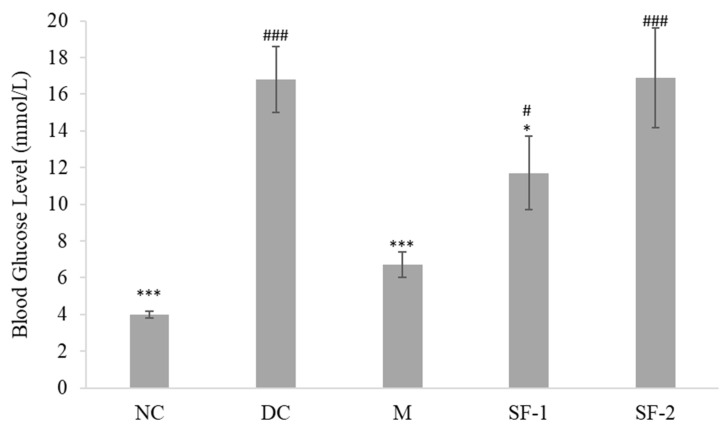
Effects of the subfractions of *S. polyanthum*‘s chloroform fraction on the blood glucose levels of streptozotocin-induced diabetic rats after being treated orally twice daily for six days. NC, normal control (normal saline, 10 mL/kg); DC, diabetic control (normal saline, 10 mL/kg); M, metformin (500 mg/kg); SF-1, subfraction-1 (250 mg/kg); SF-2, subfraction-2 (250 mg/kg). The values are expressed as mean ± SEM (*n* = 6). # indicates significant differences as compared to the NC (^###^
*p* < 0.001 and ^#^
*p* < 0.05). * indicates significant differences compared to the DC (*** *p* < 0.001 and * *p* < 0.05) as analyzed using Dunnett’s as post hoc test.

**Figure 3 molecules-27-06814-f003:**
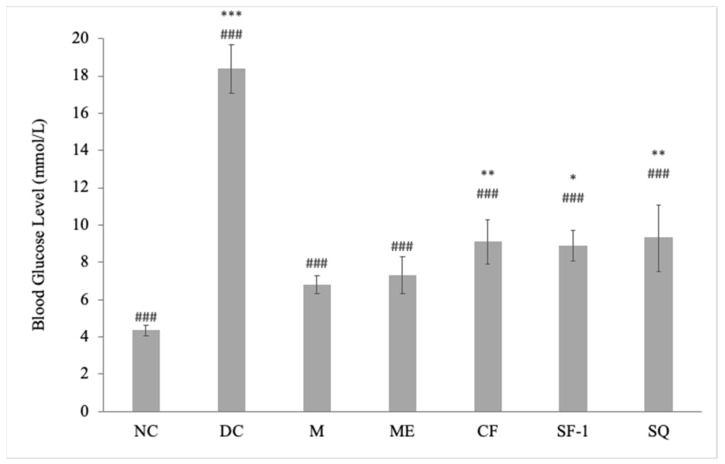
Effects of *S. polyanthum*’s active extract, fraction, and subfraction on the blood glucose levels of streptozotocin-induced diabetic rats after being treated orally twice daily for twelve days. NC, normal control (normal saline, 10 mL/kg); DC, diabetic control (normal saline, 10 mL/kg); M, metformin (500 mg/kg); ME, methanolic extract (1 g/kg); CF, chloroform fraction (500 mg/kg); SF-1, *n*-hexane fraction (250 mg/kg); SQ, squalene (160 mg/kg). The values are expressed as mean ± SEM (*n* = 6). # indicates significant differences as compared to the DC (^###^
*p* < 0.001). * indicates significant differences compared to the NC (*** *p* < 0.001, ** *p* < 0.01, and * *p* < 0.05) as analyzed using Dunnett’s as post hoc test.

**Figure 4 molecules-27-06814-f004:**
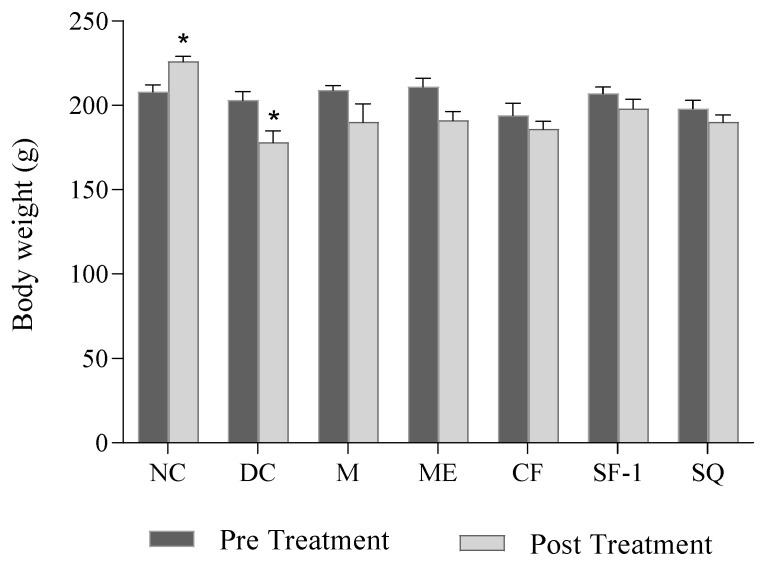
Effects of *S. polyanthum* on the body weights of streptozotocin-induced diabetic rats. All treatments were administered orally twice daily for 12 days as follows: NC, normal control (saline, 10 mL/kg); M, metformin (500 mg/kg); DC, diabetic control (saline, 10 mL/kg); ME, methanol extract (1 g/kg); CF, chloroform fraction (500 mg/kg); SF-1, *n*-hexane fraction (250 mg/kg); SQ, squalene (160 mg/kg). The values are expressed as mean ± SEM (*n* = 6). * indicates significant differences between pre-treatment (day 0) and post-treatment (day 12) of the same group as analyzed by using the paired *t*-test (* *p* < 0.05).

**Figure 5 molecules-27-06814-f005:**
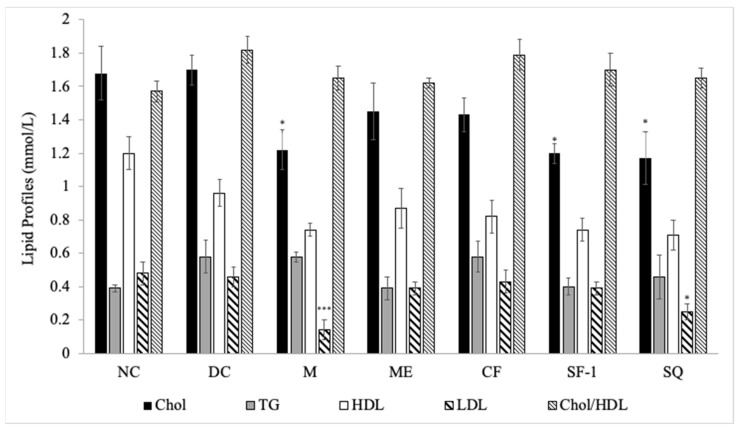
Effects of *S. polyanthum* on the lipid profiles of streptozotocin-induced diabetic rats after 12-day twice-daily oral treatments. NC, normal control (normal saline, 10 mL/kg); DC, diabetic control (normal saline, 10 mL/kg); M, metformin (500 mg/kg); ME, methanol extract (1 g/kg); CF, chloroform fraction (500 mg/kg); SF-1, subfraction-1 (250 mg/kg); SQ, squalene (160 mg/kg); Chol, cholesterol; TG, triglycerides; HDL, high-density lipoprotein; LDL, low-density lipoprotein. The values are expressed as mean ± SEM (*n* = 6); * *p <* 0.05 and *** *p <* 0.001 indicate significant differences compared to the DC as analyzed using Dunnett as post hoc test.

**Figure 6 molecules-27-06814-f006:**
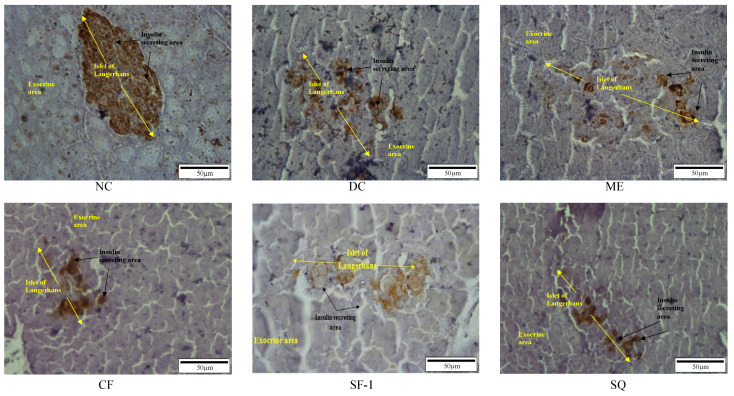
Immunohistochemistry staining of the β-cells pancreas after being treated for 12 days.

**Figure 7 molecules-27-06814-f007:**
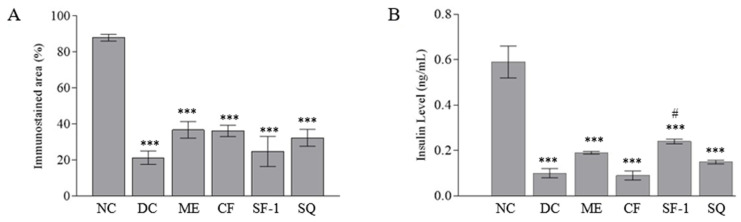
(**A**) Percentage of the immunostained area of each treated group and (**B**) Concentration of serum insulin of each treated group. NC, normal control (saline, 10 mL/kg); M, metformin (500 mg/kg); DC, diabetic control (saline, 10 mL/kg); ME, methanol extract (1 g/kg); CF, chloroform fraction (500 mg/kg); SF-1, *n*-hexane fraction (250 mg/kg); SQ, squalene (160 mg/kg). The values are expressed as mean ± SEM (*n* = 6). * indicates significant differences as compared to the NC. # indicates significant differences compared to the DC (*** *p* < 0.001 and # *p <* 0.05) as analyzed using Dunnett as post hoc test.

**Figure 8 molecules-27-06814-f008:**
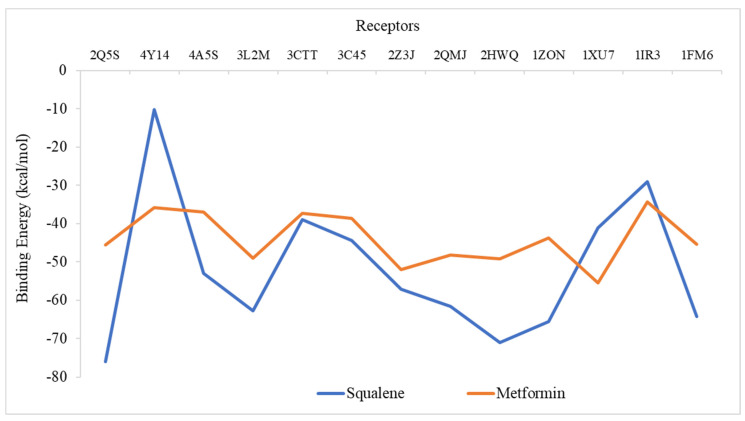
Binding energies of the ligands against thirteen protein receptors related to diabetes mellitus. 1FM9, heterodimer of the human RXR α and peroxisome proliferator–activated receptor γ ligand binding domains bound with 9–cis retinoic acid and GI262570 and co–activator peptides; 1IR3, phosphorylated insulin receptor tyrosine kinase in complex with peptide substrate and ATP analog; 1XU7, tetrameric 11B–HSD1; 1ZON, CD11A I–domain without bound cation; 2HWQ, peroxisome proliferator–activated receptor agonists; 2Q5S, peroxisome proliferator–activated receptor γ bound to partial agonist NTZDPA; 2QMJ, N–terminal subunit of human maltase–glucoamylase in complex with acarbose; 2ZJ3, isomerase domain of human glucose:fructose–6–phosphate amidotransferase; 3C45, human dipeptidyl peptidase IV/CD26 in complex with a fluoroolefin inhibitor; 3CTT, N–terminal human maltase–glucoamylase with casuarina; 3L2M, pig pancreatic alpha–amylase with alpha–cyclodextrin; 4A5S, human DPP4 in complex with a novel heterocyclic DPP4 inhibitor; 4Y14, tyrosine phosphatase 1B complexed with inhibitor.

**Figure 9 molecules-27-06814-f009:**
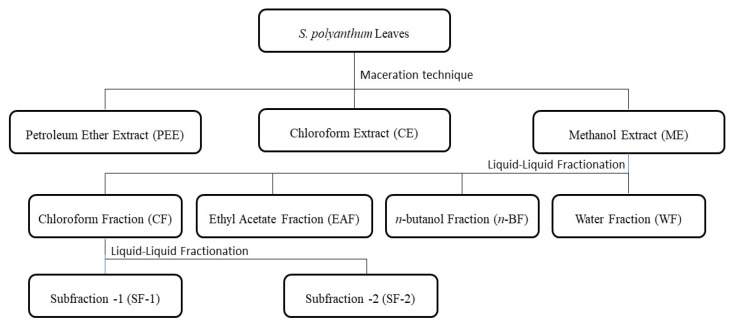
Schematic representation of bioactive-guided fractionation of *S. polyanthum* leaves.

**Figure 10 molecules-27-06814-f010:**
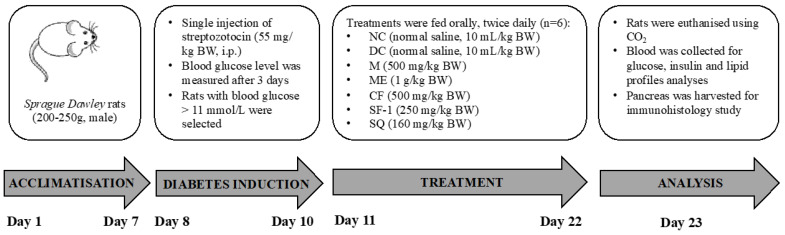
In vivo antihyperglycemic test.

**Table 1 molecules-27-06814-t001:** Phytochemical components identified in the methanolic extract (ME), chloroform fraction (CF), and *n*-hexane subfraction-1 (SF-1) using GC–MS.

Samples	RT *	Peak (%)	Compound Name	Molecular Formula
ME	14.96	7.60	Squalene	C_30_H_50_
CF	10.74	6.90	Hexadecanoic acid, methyl ester	C_17_H_34_O_2_
	11.49	2.03	9,12-Octadecadienoic acid, methyl ester	C_19_H_34_O_2_
	14.97	8.92	Squalene	C_30_H_50_
	18.37	4.73	Vitamin E	C_29_H_50_O_2_
	21.66	22.57	Stigmasterol, 22,23-dihydro-	C_29_H_50_O
SF-1	14.92	4.54	Squalene	C_30_H_50_
	18.28	4.26	Vitamin E	C_29_H_50_O_2_
	21.50	33.37	Stigmasterol, 22,23-dihydro-	C_29_H_50_O

* RT = retention time; Peak area = the percentage of the compound it represents.

## Data Availability

Not applicable.

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
