# Peer review of "Bioactivity-Guided Fractionation and Identification of Antidiabetic Compound of Syzygium polyanthum (Wight.)’s Leaf Extract in Streptozotocin-Induced Diabetic Rat Model"

_molecules, 2022, doi:10.3390/molecules27206814_

Round 1

Reviewer 1 Report (New Reviewer)

Widyawati et al. present the results of a study in diabetic rats assessing the effects of fractions and subfractions of S. polyanthum, comparing to squalene. This is a nice study with appropriate end-points, however I have a few comments regarding clarity and also rigor of data analysis and therefore the conclusions made.

Please make clear throughout whether discussing type 1 or type 2 diabetes.   The introduction seems to talk about both, especially since the increasing prevalence in mentioned.  The study appears to be carried out on rats modelling type 1 diabetes (as STZ is used alone, not in combination with NA), which is confirmed with the pancreatic IHC showing reduced % coverage of β cells.

Line 59, “the author’s preliminary studies” sounds like a couple of experiments carried out at the beginning of this study, if they are referring to the paper cited, consider “Our previous study”.

Throughout the study I have major concerns about the statistical tests employed – these should not be t-test, unless the effect of each fraction is shown in it’s own graph. Please consult a statistician.  Further to this, statements such as “…exerted a stronger hypoglycemic effect…” (ln 93) and “…magnitude of the hypoglycemic effect decreased….” (ln129) cannot be made just by looking at the averages – stats need to be carried out comparing the effect of the different fractions to each other, such as also showing the results as % change from their respective baselines.

In figure 3, were the post-treatment levels compared to the NC group? It would be interesting to see if any fractions statistically eliminate the hyperglycemia, in addition to reducing it from its own baseline.

Line 161 – NC rats increased in body weight – please show this on the graph. However, this also shouldn’t be t-tests, unless separated into separate graphs for each treatment.

Which tests were used for table 2 and figure 5?

Although explained in the previous section, please explain a little further the point made on lines 228-230.

Author Response

Reviewer 2 Report (New Reviewer)

The manuscript talks about the hypoglycemic effect of different extracts and subfractions of Syzygium polyanthum.

1.-indicate in the methodology the type of diabetes that was induced in the rats

2.include in section  "2.2 Effect on blood glucose level", a normoglycemic control (figure 1 and 2).

Author Response

Dear Editor,

Dear Reviewer,

Thank you for the insightful comments on the paper. We have modified the manuscript accordingly. Here is a point-to-point response to the reviewers’ comments and concerns.

1. Indicate in the methodology the type of diabetes that was induced in the rats

The type of diabetes that was induced in the rats has been indicated by revising the subheading of Section 3.5.1 into "Induction of type 1 diabetes mellitus in rats.

2. Include in the section "2.2 Effect on blood glucose level", a normoglycemic control (figures 1 and 2).

The figures have been revised by adding the normal control (NC) group, as suggested.

Thank you.

Reviewer 3 Report (New Reviewer)

In this paper by Widyawati et al. entitled "Bioactivity-Guided Study of the Hypoglycemic Properties of a Squalene-Rich Extract of Syzygium polyanthum (Wight.) in Streptozotocin-Induced Diabetic Rat Model", the authors establish the hypoglycemic activity of S. polyanthum leaf extract. They indicate the hypoglycemic effect of methanol extract of S. polyanthum against diabetic mellitus. They suggest that squalene is one of the candidate molecules that have a bioreactivity of S. polyanthum. These observations are interesting; however, I have the following comments:

Major comments:

1. The authors measured several parameters including body weight, blood glucose concentrations, lipid profiles and insulin levels. They are important observations. However, the data presented in this manuscript do not advance the understanding of molecular mechanism of how S. polyanthum leaf extract exerts its physiological function. The authors should perform additional experiments to clarify the molecular mechanism.

2. Table 2: The authors should indicate significant differences as compared to the DC.

3. Lines 185-194 and Table 2: ME (7.6% squalene), SF-1 (4.54% squalene) successfully decrease the levels of TG in the diabetic rats. In contrast, CF (8.92% squalene) failed to decrease the TG level. The standard error of TG of SQ treatment is large. I don’t think the squalene is an important factor of S. polyanthum leaf extract. The authors should present experimental evidence and indicate the scientific significance of the treatment of squalene-rich extract to the diabetic animals.

4. Fig. 3: The authors should indicate the time course of blood glucose levels (from day 0 to day 12). It will be helpful for readers to understand the pharmacological effects of S. polyanthum leaf extract.

5. Fig. 5: The authors suggest that insulin is NOT related to the hypoglycemic effect of squalene-rich extract of S. polyanthum. They also report that squalene do not restore the insulin level. The authors should investigate the molecular mechanism of the hypoglycemic effect of squalene-rich extract of S. polyanthum. The authors should provide a more mechanistic interpretation.

6. How did the authors determine the dosage? Have the authors checked the safety or toxicity? I think the experiments in this manuscript lacks pharmacological validation.

7. Fig. 5A-ME, CF, SQ: The images are out of focus.

Minor comments:

1. Introduction: A detailed introduction on the pharmacology and physiology of squalene should be provided. The authors cite only their own work [11].

2. Line 240, and 355: The authors should describe the name of the sample in which insulin was measured. Plasma? Serum?

3. Fig. 5A: The letters indicating the length of the scale bar should be written larger. 

4. Table 2: I think it is difficult for readers to understand the contents of the table. Table 2 should be converted into graphs. In addition, the authors need to clearly state statistically significant differences. (It would be better to move Table 2 itself into the supplemental data to refer to the measurements.)

5. Line 332: When did the authors inject streptozotocin into the rats and start the oral administration of extract(s)? It is difficult for readers to understand the experimental time course.

6. Line 356-359: The authors should provide more detailed information on how the lipids are analyzed.

Round 2

Reviewer 1 Report (New Reviewer)

The authors have made all changes requested resulting in an improved manuscript. One small further change to improve the abstract is suggested - "(type 1 diabetes model)" to be added following "Streptozotocin-induced diabetic rats."

Author Response

The authors have made all changes requested resulting in an improved manuscript. One small further change to improve the abstract is suggested - "(type 1 diabetes model)" to be added following "Streptozotocin-induced diabetic rats."

Appreciate the positive comment. We have amended the sentence in the Abstract, as follows:

"Streptozotocin-induced type 1 diabetic rat was used ..."

Thank you.

Reviewer 2 Report (New Reviewer)

The authors made the suggested corrections, however I have two additional comments that came up when reviewing the latest version.

1. In the foot of figure 2 indicate what NC means

2. Check that the word in vitro is in italics throughout the document

Author Response

Dear Reviewer,

We appreciate your comments and suggestion. Amendments have been made accordingly, as follows:

  1. We have defined the abbreviation of NC in the legends of Figure 1 and 2, as follows: NC, normal control (normal saline, 10 mL/kg).
  2. We have carefully gone through the manuscript and revised the word in vitro.

Thank you.

Reviewer 3 Report (New Reviewer)

The authors modified the manuscript properly, and responded to my comments. I have the following comments:

Minor comments:

1) Fig. 6: “50 µm” is too small. Please use larger fonts.

2) Fig. 8: The authors should provide more detailed explanations in the legends. I think the readers will have difficulty understanding what the horizontal axis represents. I think the explanations described in the Materials and Methods section (lines 435-448) should be included in the legends to Fig. 8. 

ex.) 2Q5S, PPARGAMMA bound to partial agonist NTZDPA; 4Y14, Tyrosine phosphatase 1B complexed with inhibitor; 4A5S, ….

3) Lines 320 & 326: “Figure 8” should be “Figure 9”.

4) Lines 371 & 386: “Figure 9” should be “Figure 10”.

Please check the revised manuscript carefully.

Author Response

Dear Reviewer,

We appreciate the constructive comments and suggestions. We have amended the manuscript accordingly, as follows:

1) Fig. 6: “50 µm” is too small. Please use larger fonts.

The font has been enlarged and the figure has been replaced.

2) Fig. 8: The authors should provide more detailed explanations in the legends. I think the readers will have difficulty understanding what the horizontal axis represents. I think the explanations described in the Materials and Methods section (lines 435-448) should be included in the legends to Fig. 8. ex. 2Q5S, PPARGAMMA bound to partial agonist NTZDPA; 4Y14, Tyrosine phosphatase 1B complexed with inhibitor; 4A5S, ….

Thank you for the suggestion. We have included a details explanation of the protein receptors in the legend for Figure 8. 

3) Lines 320 & 326: “Figure 8” should be “Figure 9”.

Thank you. Amendments have been made accordingly.

4) Lines 371 & 386: “Figure 9” should be “Figure 10”.

Amendments have been made accordingly. Thank you.

This manuscript is a resubmission of an earlier submission. The following is a list of the peer review reports and author responses from that submission.

Round 1

Reviewer 1 Report

Overall the quality of this manuscript is very poor. 

  1. The introduction does not adequately introduce the topic.
  2. Make clear in the introduction that immunogenic destruction of the beta cell is associated with type 1 diabetes, not type 2.
  3. The figure numbers and labels do not match-up with what is described in the text. This makes it impossible to follow and so therefore difficult to assess what has been done.
  4. The section on weight loss is muddled and largely irrelevant.
  5. There are no sections in figure 2 but text says sections a and c. What is this supposed to be?
  6. Figure 3: it is impossible to read the scale bar. How many animals were used in total in this analysis? The statistics are not accurately shown and are not described at all in the figure legend. 
  7. The data does not support the conclusions.

Reviewer 2 Report

The manuscript entitled: „ Bioactivity-Guided Study of Antihyperglycemic Effect of 2 Syzygium polyanthum (Wight) Walp. Extracts in Streptozotocin Induced Diabetic Rats” describe effect of 2 different exptracts from syzygium polyanthum on glucose level in STZ rats. Although the manuscript is very basic, it seems to be partially interesting, however it contains many errors and omissions. I would just like to point out some major corrections that will improve your understanding of the article and perhaps help to publish it after major corrections and re-submission.

Major

  • The optimal temperature for laboratory rat is 21-22 o Why the Authors kept the animals at such a high temperature (25-30 oC)
  • Why the Authors used NaCl for STZ dissolving? a more standard buffer is the citrate buffer
  • If the Authors have measured glucose and insulin concentration it may be worth to add calculations for insulin sensitivity or insulin resistance indices (HOMA-IR, Quicki etc).
  • “The observed antihyperglycemic effect of S. polyanthum could be attributable to other mechanisms of action involving extra-pancreatic pathways which include inhibition of dipeptidyl-peptidase-IV (DPP-IV) or glucagon-like peptide-1 receptor (GLP-1R) agonist, suppression of intestinal glucose absorption and stimulation of glucose uptake by liver, muscle and adipose tissues.” - How the Authors made this conclusion since they did not investigate any of these metabolic pathways
  • Why did the authors apply two-factor analysis of variance to all statistical calculations? Even to the results of lipid proliferation or glucose / insulin concentration – please explain.

Minor

  • Why the Authors measured blood glucose only using a glucometer? It is allowed for in vivo measurements, but for measurements after the end of the experiment, they should use other, more accurate methods, e.g. enzymatic methods.
  • Figures descriptions The description of figure 1 does not include the description of individual figures A, B, C
  • Due to the fact that I am not a native speaker and I cannot say the correctness of English, it seems to me that the language of paper needs to be improved by a professional language company or native English speaking person.
  • Please add an index of abbreviations, which will make it easier for the reader to understand the text, without looking for their development in the text.